# A Standardised Core Outcome Set for Measurement and Reporting Sedentary Behaviour Interventional Research: The CROSBI Consensus Study

**DOI:** 10.3390/ijerph19159666

**Published:** 2022-08-05

**Authors:** Fiona Curran, Kieran P. Dowd, Casey L. Peiris, Hidde P. van der Ploeg, Mark S. Tremblay, Grainne O’Donoghue

**Affiliations:** 1School of Public Health, Physiotherapy and Sports Science, University College Dublin, D04 V1W8 Dublin, Ireland; 2Department of Sport and Health Sciences, Technological University of Shannon, N37 HD68 Athlone, Ireland; 3Department of Physiotherapy, School of Allied Health, Human Services and Sport, La Trobe University, Melbourne 3086, Australia; 4Amsterdam UMC, Department of Public and Occupational Health, Amsterdam Public Health Research Institute, Vrije Universiteit Amsterdam, van der Boechorststraat 7, 1081 BT Amsterdam, The Netherlands; 5Children’s Hospital of Eastern Ontario Research Institute, Ottawa, ON K1H 8L1, Canada; 6Department of Pediatrics, University of Ottawa, Ottawa, ON K1N 6N5, Canada; 7Department of Health Sciences, Carleton University, Ottawa, ON K1S 5B6, Canada

**Keywords:** sedentary time, stationary behaviour, physical inactivity, minimum dataset, Delphi, COS

## Abstract

Heterogeneity of descriptors and outcomes measured and reported in sedentary behaviour (SB) research hinder the meta-analysis of data and accumulation of evidence. The objective of the Core Research Outcomes for Sedentary Behaviour Interventions (CROSBI) consensus study was to identify and validate, a core outcome set (COS) to report (what, how, when to measure) in interventional sedentary behaviour studies. Outcomes, extracted from a systematic literature review, were categorized into domains and data items (COS v0.0). International experts (n = 5) provided feedback and identified additional items, which were incorporated into COS v0.1. A two round online Delphi survey was conducted to seek consensus from a wider stakeholder group and outcomes that achieved consensus in the second round COS (v0.2), were ratified by the expert panel. The final COS (v1.0) contains 53 data items across 12 domains, relating to demographics, device details, wear-time criteria, wear-time measures, posture-related measures, sedentary breaks, sedentary bouts and physical activity. Notably, results indicate that sedentary behaviour outcomes should be measured by devices that include an inclinometry or postural function. The proposed standardised COS is available openly to enhance the accumulation of pooled evidence in future sedentary behaviour intervention research and practice.

## 1. Introduction

Sedentary behaviour (SB), defined as an energy expenditure ≤1.5 metabolic equivalents (METs) in a sitting, reclining or lying posture while awake [1], is associated with poor health outcomes, morbidity and all-cause mortality [2,3,4,5]. The recently updated physical activity guidelines of the World Health Organisation include a strong recommendation to limit the amount of time in SB across the lifespan and to replace it with any type of physical activity [6]. Internationally, this recommendation is increasingly incorporated into public health guidelines and policy [7,8,9,10].

Despite these recommendations and recent SB research, pooled evidence from reviews on the effectiveness of interventions to reduce SB is limited by heterogeneity of outcomes [11,12]. This field of research spans many disciplines, interventions and populations, and the heterogeneity of outcomes, ‘what to measure and report’, is a fundamental stumbling block to the accumulation and aggregation of evidence [13,14,15]. Rapid growth in both laboratory and personal activity sensing devices now provide an abundance of data for potential analysis [16,17,18,19]. The ability to gather activity data has forged ahead of the human ability to easily understand the relevance and importance of that data. Critically, the data are neither standardised nor interoperable, arising from a broad range of devices and software which quantify SB and activity differently while also providing a multitude of diversely quantifiable variables [20,21,22,23,24,25,26,27,28]. In this data-rich but information-poor environment [29], what to measure and report is challenging, often arbitrary and ineffective for research synthesis.

Minimum datasets provide data which allows like-for-like comparisons and are the foundation for standardised interoperable clinical data and electronic health records. They have been developed and incorporated into clinical practice for decades [30]. A minimum dataset consists of a defined set of data items that are considered the minimum essential components required to meet a particular purpose. This term has largely been replaced in interventional research by ‘core outcome set’ (COS), defined as an agreed standardised minimum set of core outcomes that should be reported in all studies, but which is flexible enough to allow additional measures to be incorporated [31]. Development and use of COS’s align to the latter two principles of findable, accessible, interoperable and reusable (FAIR) data [32].

Consensus and standardisation of outcomes, i.e., project descriptors and reported measures is necessary to improve the synthesis of findings from future SB interventional studies, and to harmonise methods, develop FAIR data and develop standardised SB surveillance systems [13,33]. This research is timely, following the SB terminology consensus project, which provided clear standardised language for examining patterns of movement behaviour across all domains of living [1]. The SB consensus terminology framework provides a foundation for standardisation of measurement and reporting of SB, enabling the development of a core outcome set (COS), thus enhancing research, and ultimately health outcomes, related to SB.

The development of a standardised COS for reporting SB outcomes will facilitate comparability and meta-analyses of data from intervention trials in this complex research field. This critical action research project was identified following a recent systematic literature review, led by members of the research team, of randomised controlled trials of interventions designed specifically to change SB across all domains of living, unpublished and published [15]. Consistent with a prior review [34], heterogeneity in measuring and reporting SB in interventions was a limitation noted in the review, preventing inclusion of all of the study data in meta-analyses.

Therefore, the objective of this study was to identify and validate by consensus, a core set of domains and items (‘what to measure’, ‘how to measure’, ‘when to measure’, ‘what to report’) for device measured SB in intervention studies of adult populations.

## 2. Materials and Methods

### 2.1. Study Design

University College Dublin (UCD) human research ethics committee reviewed the study and granted low risk ethical approval (reference number: LS-E-20-157). The protocol and study was guided by, and registered on, the Core Outcome Measures for Effectiveness Trials (COMET) database (https://www.comet-initiative.org/Studies/Details/1897 accessed on 22 June 2022) and reported using the Core Outcome Set-STAndards for Reporting (COS-STAR) statement [35]. A pragmatic, six step bottom–up approach, similar to that used in prior studies [36,37], and a modified nominal Delphi method for consensus [38] was employed (Figure 1).

### 2.2. Outcome Identification

A recent PROSPERO registered (CRD42020172457) systematic literature review, conducted by members of the research team, was used to identify outcomes from randomised controlled trials (RCTs) of interventions targeting SB change in ambulatory adults [39]. The systematic review aimed to determine the effect of interventions to reduce SB, thus, retrieved studies were appropriate to identify a comprehensive list of potential outcomes. The methodology is available in the PROSPERO protocol and the published review of a subset of the studies [15].

In brief, six databases were searched from January 2000–December 2020. Inclusion criteria were randomised controlled trials in adult populations (age 18–65), where change in SB was the primary outcome. Studies of workplace and non-workplace interventions were included, while those including children, older adults or clinical populations were excluded. A total of 5589 de-duplicated studies were identified, of which 22 were finally included (flowchart in Appendix A).

Data were extracted for identification of potential core descriptors and outcomes, from studies of interventions across all domains of living (n = 22). The following data were extracted from each study: study type; author details; year and journal of publication; population demographics; devices used; criteria for inclusion in analysis; outcomes measured; method of measurement; and statistics reported.

Descriptors and outcomes were identified from the methods and results section of each paper. Each represents one data item and groups of similar data items were categorised into appropriate data domains. All identified data items were included in the first iteration for review by experts

### 2.3. Stakeholder Identification and Expert Review

The principal investigator invited five experts, each with multiple relevant publications in peer reviewed journals, to participate on the expert panel. The expert panel function was to review and recommend changes to the draft COS v 0.0, to identify additional items, review any additional items identified by participants and ratify the final COS.

An initial list of all identified descriptors and outcomes, data items, from the systematic literature review was tabulated in an Excel spreadsheet, and grouped into data domains, with key terms explained. The expert group was asked to identify additional, or missing items that they thought should be included, and if any of the identified items should be excluded. Each expert also reviewed the list of data items and domains for comprehension and suitability. Feedback was discussed by the research team and incorporated into round 1 of the Delphi survey. All additional outcomes identified by the experts were also added to the list of data items to be used in round 1 of the Delphi. An online survey was developed. Each domain was constructed as a single question containing multiple data items, for participants to rate on a 9-point Likert scale, from “not important” to “critical”.

A larger stakeholder group is required to provide further development and wider consensus validation of a potential COS. The Sedentary Behaviour Research Network (SBRN), which is a global organization that focuses specifically on the health impact of SB was identified as a suitable group. Members consist of researchers from multiple disciplines, academics and health professionals (n > 500 active members at time of Delphi).

### 2.4. Consensus Process Delphi Survey

DelphiManager, an online survey system managed and maintained by the COMET initiative, was used to distribute and manage the Delphi process (https://www.comet-initiative.org/delphimanager/ accessed on 22 June 2022). The objectives and details of the study were presented at the beginning of each round. Two online rounds were completed. To minimise attrition, participants were informed of the importance of completing the entire Delphi process and received up to three reminder emails and clear timelines to complete each round.

Potential participants viewed an information page about the study before registering their consent and subsequently provided demographic data [age, gender, country of practice, current role (clinical, academic, research), and years of experience]. Participants remained anonymous by use of a unique identifier and all data were stored against the unique identifier only, preventing identification of participants or individual responses.

Participants were asked to rate each of the data items using the Grading of Recommendations Assessment, Development and Evaluations (GRADE) scale, a nine-point Likert scale, with 1 to 3 labelled ‘not important’, 4 to 6 ‘important but not critical’, and 7 to 9 ‘critical’ [40]. Participants could also select ‘unable to score’ and comment on any individual item, make general comments at the end of the survey or suggest additional data items.

### 2.5. Delphi Survey Round One: Expert Panel Snowball and SBRN

Members of the SBRN were notified of the study in two consecutive monthly e-newsletters (June/July 2021) and were asked to complete a Delphi survey via an embedded link. Additionally, to ensure uptake of the consensus study, the expert panel ‘snowballed’ invitations to participate through their networks. Snowball sampling is a technique to identify potential subjects for studies where subjects are hard to locate [41]. The total number of invitations was thus not available, but the number of participants recruited and those completing each round were recorded.

The results were analysed to identify missing data, generate the distribution of scores and calculate the median scores. The research team reviewed and agreed on any additional items for inclusion in round 2.

### 2.6. Delphi Survey Round Two

Only participants who completed round 1 were invited to complete round 2. Participants were informed of the overall consensus scores for each data item retained from round 1 and were reminded of their own round 1 score via the Delphi manager. Each participant was again asked to rate each retained data item and the additional items from round 1.

### 2.7. Retaining or Dropping Items between Rounds

When the initial list of outcomes is large, as in this case, including them all in each Delphi round may impose a sufficient burden on participants that it results in considerable attrition from one round to the next, but dropping items between rounds risks excluding important outcomes [31,42]. Therefore, as suggested in the COMET handbook, we pre-defined broader criteria for retention and dropping of outcomes between rounds, to reduce the likelihood that important outcomes were dropped [31,40,41]. Outcomes were retained between rounds if ≥50% participants scored 7 to 9 and ≤15% scored 1 to 3 on the Likert scale, and outcomes were dropped between rounds if ≥50% participants scored 1 to 3 and ≤15% scored 7 to 9 on the Likert scale.

### 2.8. Consensus for Inclusion, Definition

Consensus threshold criteria for inclusion or exclusion in the final COS was defined a priori. Consensus to include an outcome was achieved if ≥70% of participants rated the item’s importance from 7 to 9 (critical) and ≤15% rated it from 1 to 3 (not important) on the Likert scale. Conversely, consensus to exclude an outcome was achieved if ≥70% of all participants rated the item from 1 to 3 (not important) and ≤15% rated the item’s importance from 7 to 9 (critical) on the Likert scale [31]. Items not reaching these thresholds are reported as “no consensus reached”.

## 3. Results

### 3.1. Delphi Survey Participants’ Characteristics

A total of 59 participants from 15 countries registered for the Delphi survey and 50 (85%) completed round 1. The majority of participants were female (58%), in an academic role (68%) and had >5 years’ experience in SB research (60%). Forty-one participants who completed round 1 also completed round 2 (response rate 82%), of whom 68% had >5 years’ experience in SB research (Table 1).

### 3.2. Outcomes Identified (Cos V.0 to Cos V.1)

One-hundred and ten items were initially identified from the literature and categorised into 19 domains (COS v 0.0) (Appendix A). Following review by the expert panel, eight additional items were identified (Appendix A). Further discussion led to the exclusion of some domains and items that were deemed ‘non-core outcomes’ because they were specific to a particular field of research (e.g., biomarkers). A number of items were merged or re-categorised into different domains. Ninety-four data items across 13 domains (COS v 0.1) were used to construct round 1 of the Delphi survey (Appendix A).

### 3.3. Delphi Survey Analysis Round 1

Following the round 1 Delphi survey, twenty-five items were deemed ‘obvious’ and retained for the next version of the COS for ratification by the expert panel (v 0.2). Ten of the obvious items related to ‘when to measure’, eleven to ‘the reporting statistic’ and the final four items comprising the domain of ‘outcomes related to wear-time criteria’ were also deemed obvious. To reduce participant burden and associated attrition [42], these obvious items, which all achieved consensus (range 79–100%), were not included in round 2 of the survey.

Based on pre-defined criteria, seventeen items and one domain (reporting energy expenditure) did not meet the consensus criteria to retain and were dropped at this stage. Fourteen additional items were suggested by participants, three of which were modified and added in round 2 (Appendix A). Fifty-two items met the criteria for retention between rounds, resulting in fifty-five items for participants to rate in the second round. Ratings and inclusion decisions are provided in Appendix A.

### 3.4. Delphi Survey Analysis Round 2

Twenty-nine of the fifty-five data items rated by participants in round 2 of the Delphi met the consensus criteria for inclusion in the final COS. Participants were also asked to provide a reason if their rating changed from round 1 to round 2 and crossed a consensus boundary. Not all participants provided reasons for changing rating, but common reasons reported by those who did (n = 16) included; reconsidered based on others consensus and items not being core for all SB research. Ratings and inclusion decisions for round 2 are available in Appendix A.

### 3.5. Expert Panel Ratification (COS V 0.2 to COS V 1.0)

At the ratification meeting, the expert panel reviewed the 54 data items, comprised of the 25 obvious items from round 1 and the 29 items that met the consensus criteria in round 2 of the Delphi survey. Following discussion, a total of 53 items were ratified. Regarding type of device to measure SB, ‘inclinometer’ was not ratified by the expert panel, and ‘combined inclinometer/accelerometer’ was modified to ‘accelerometer with an inclinometer function’ to be more precise (see discussion). Experts also agreed that in the demographic domain, age, gender and population type do not need to be reported at follow-up. The list of all 53 items remaining in the ratified COS v 1.0, is available in Appendix A. Notably, 21 items are repetitions of ‘measure at baseline’, ‘measure at follow up’ or the reporting statistics. The outcome identification and consensus flow is illustrated in Figure 2.

The final COS, in user friendly format, is presented in Table 2. It is also available in open-access format for widespread adoption in all SB interventional studies via the SBRN website (www.sedentarybehaviour.org).

## 4. Discussion

This study developed a COS to guide measurement and reporting in SB intervention research, using a pragmatic, modified nominal Delphi method for consensus, inclusive of a broad international network of researchers and expert panellists. COS development is an iterative, multistage process that begins with ‘what to measure’, with subsequent development of ‘how to measure’, ‘when to measure’ and ‘what to report’ [31,43]. This study successfully answers these questions for SB intervention research and provides a template for advancing and continuing development in this field of research (Table 2 and Appendix A). Data items and domains are related to demographics, device details, wear-time criteria, wear-time measures, posture-related measures, sedentary breaks, sedentary bouts and physical activity.

Similar developments exist in other research fields [44,45,46] and although some recommendations exist for various SB measurement tools and questionnaires [22,47], no other COS exists for device-measured SB to our knowledge. This tool is recommended for use in all studies where change in SB is an outcome. It should not limit the measurement or reporting of outcomes, rather be used in conjunction with other research outcomes or other COS’s specific to a particular research question.

With regard to the type of device to measure SB, two device types met the consensus inclusion criteria, inclinometer and combined accelerometer/inclinometer. However, following expert panel discussion at the ratification meeting, there was agreement that few devices actually use true inclinometry or can be classified as an inclinometer alone, but rather, are accelerometers with an ‘inclinometer function’. A number of participants made similar comments during the Delphi survey, and, notably, accelerometry alone as a measurement device came close to, but did not achieve, consensus (67% rated 7–9 on Likert scale). This highlights the variability in research practice and terminology, the dilemma faced in measuring and reporting and the need for further clarity.

An accelerometer measures the frequency and amplitude of acceleration (counts) of the body in three planes with the periods of non-movement interpreted as SB, determined by an intrinsic count threshold [43,48]. As such, stationary standing can be misclassified as SB, and active versus passive sitting, reclining or lying is not distinguishable by accelerometry alone. By contrast, a device with an inclinometry function seeks to distinguish standing, sitting/lying or sleep/non-wear by combining a gravitational component with accelerometry and intrinsic data-processing algorithms [21,27]. The Delphi survey results identify that device-measured SB must include a postural component. This is aligned with the consensus definition of SB [1], which includes sitting, reclining and lying, and confirms the need to differentiate sitting from standing and reclined postures in the measurement of sedentary time [23,49]. The expert panel agreed that the term inclinometer is somewhat misleading, and that accelerometry is essential to the measurement of SB. Therefore, regarding the type of device to measure SB, ‘inclinometer’ was not ratified by the expert panel, and ‘combined inclinometer/accelerometer’ was modified to ‘accelerometer with an inclinometer function’. This is a key recommendation that describes critical functionality of devices (i.e., inclusion of posture and distinction between sitting and standing) for the accurate measurement and quantification of SB in interventions that aim to modify this behaviour.

Within the domain of posture-related outcomes, most interestingly, both sedentary time and sitting time achieved consensus for inclusion in the COS. This reflects the research drive to differentiate these measures because sitting time is a component of, but not necessarily equivalent to, sedentary time. Isolating sitting time from sedentary time will allow the identification of time spent in reclining and lying [23,49]. This in turn may provide new insight regarding the health impact of each of these postures. Furthermore, sitting time may be passive or active, exceeding the threshold of ≤1.5 METs energy expenditure, which defines SB (e.g., working on an assembly line while seated). The isolation of specific behaviours within sitting postures (e.g., transportation time, cycling), will advance our understanding of SB and its relationship with health. The importance of developing devices that accurately measure sedentary time, by including both the postural and energy components of SB, was highlighted in recent SB guidelines [50,51]. Emergent methods reported to date to synchronously measure both components of SB, include proprietary algorithms with single or multiple sensors [51,52,53,54]. The future development of devices to measure SB should continue to advance accuracy by including postural and energy components.

The measurement and reporting of sedentary breaks include both ‘sit to stand transitions’ and ‘movement breaks’, highlighting the need to differentiate breaks that include movement from those that do not, to enhance understanding of the independent impact of each. Although breaking sitting time by standing improves glycaemic control, cardio-metabolic markers and blood pressure in the short term [55,56], replacing sitting with stepping is recommended to meaningfully affect energy expenditure, waist circumference and body mass index (BMI) [57]. Movement breaks are likely to better impact health [58], and reporting these breaks will advance our understanding and inform future intervention design and recommendations to alleviate the negative health consequences of prolonged sedentary behaviour.

Regarding sedentary bouts, there is little evidence to identify critical cut-points for impacting health outcomes [59], and the length of sedentary bouts have to some degree been arbitrarily chosen by researchers. The consensus to include bouts > 30 min and >60 min are a minimum starting point to support future meta-analysis and the accumulation of evidence of the impact of prolonged bouts of SB. Although not considered ‘critical’ to all SB research, reporting outcomes related to bouts of differing length is likely to enhance understanding of the temporal relationship of sedentary bouts with health outcomes [60].

The inclusion of physical activity as a core domain, in SB research, requiring measurement and reporting of all levels of physical activity, highlights the intricate relationship of SB and physical activity in relation to health. Although targeting interventions to reduce SB, an increase in physical activity may be desired or expected. Examining and understanding the changes achieved in physical activity when reducing SB (i.e., increasing light, moderate or vigorous PA) is critical, and the potential implications on health may be substantial across the intensities of PA. Pooled analysis of the data collected from studies using the developed COS will enhance knowledge of the mediation and dependence of these outcomes and ultimately inform the development of more effective interventions.

Regarding demographic core descriptors, many participants commented that an outcome related to socioeconomic status, such as occupation, employment or education, is needed, but consensus was not achieved for any of these. However, >90% of participants rated each of these outcomes ‘important but not critical’ or ‘critical’. This finding is aligned with the definition and use of a COS (i.e., suitable for use in all research identified in the COS development but flexible enough to incorporate other outcomes) [31]. Applied to SB research, some outcomes, such as occupation, employment or education, might be considered critical if the intervention is being conducted in one of these settings. Therefore, it is essential that settings-based information, in line with best practise reporting of interventional research, should be collected and reported. Furthermore, BMI is included as a core outcome in the COS, but notably, weight and height were not considered core for measuring or reporting. This allows flexibility in measurement and reporting and may allow inclusion of self-reported measures. Researchers should continue to report items that were not considered core items for all research but that are relevant or even essential to their methodology, research question and population.

A number of participants with >10 years’ experience in SB research commented that self-report measures or daily logs are needed to validate the device measures or provide contextual understanding. Whilst beyond the scope of this study to develop a COS for self–report measures of SB, when asked as an additional item in round 2, 63% of participants considered the development and future inclusion of standardised self-report measures as ‘critical’ and a further 31% deemed this ‘important but not critical’. Self-report outcomes may be important for the confirmation of non-wear periods, sleep periods or active periods and may provide important information for some research contexts (i.e., what SB are being engaged in), questions (e.g., who is SB being accumulated with) and settings (e.g., where is SB being accumulated) [22]. Further studies to develop core self-report outcomes to accompany the COS are likely to enhance the contextual understanding of SB and its mediators in future interventional research.

Finally, core outcome sets, such as the one developed in this study, are the foundational framework for standardised interoperable datasets in health informatics technology, including health records and registries. Mapping this COS to standardised information exchange models will enable its use in public health registries and electronic health records, thereby enhancing the evidence base for the effectiveness of interventions in clinical and population health studies regarding SB.

## 5. Strengths and Limitations

This study was registered on the COMET database and followed the COS-STAR guidelines [35]. We successfully recruited and retained an international group of experienced SB researchers, in sufficient numbers [42,45,49] and maintained a good response rate [31,42]. We developed a protocol and followed a robust methodology. The Delphi survey provided anonymity and ensured equal weighting to all participants while allowing reflection on the overall consensus. The COS is openly available (www.sedentarybehaviour.org) for uptake by all stakeholders.

A representative stakeholder group was involved in the consensus process but a limitation to the study is the absence of public patient involvement (PPI). This was an accepted limitation at the outset for pragmatic and research reasons. The field of SB research spans many patient and public health domains, and choosing one population would potentially limit the generalisability of the COS. The objective was to identify a COS for use in all studies where SB is an outcome. It is our hope that PPI consensus will be included in the future, with specific populations and in specific contexts, as the uptake of the COS occurs.

Stakeholders from industry are also notably absent from the consensus process. This is both a strength, since conflicts of interest may occur (e.g., with proprietary software), and a limitation [31], since technology developers and device manufacturers can contribute to SB research by advancing the ability of devices to distinguish sedentary postures and patterns. This COS is not intended to limit current or future practice in measurement but rather to be inclusive of the rapid advances in hardware, software and processing technology, by providing a foundation for the analysis of essential outcomes.

Although we asked our experts to specifically target invitations to participants in low- and middle-income countries and we achieved reasonable diversity of international participation, low- and middle-income countries were underrepresented. This may limit the cross-cultural or contextual application of the CROSBI.

Finally, many participants expressed their appreciation of the value of the study and the need for this COS. However, the participant burden was significant and a consensus meeting was not part of the design of this study for pragmatic reasons (time and finance). It is likely that future discourse and agreement will lead to further iterations and development, particularly as technological advancements progress in the field of research.

## 6. Conclusions

In conclusion, we have developed a COS to guide measurement and reporting of descriptors and outcomes in SB intervention research in adult populations, using a robust methodology and a diverse international stakeholder group. This open access CROSBI provides a data collection and reporting tool for use across many adult populations and domains of living. Widespread adoption will improve the quality and synthesis of data from intervention studies in adults, and thereby improve synthesis of evidence, management of resources and ultimately interventions for sedentary behaviour change. Researchers should refer to this COS when designing and reporting intervention studies related to SB to improve quality, aid interpretation and allow pooling of results in meta-analyses.

## Figures and Tables

**Figure 1 ijerph-19-09666-f001:**
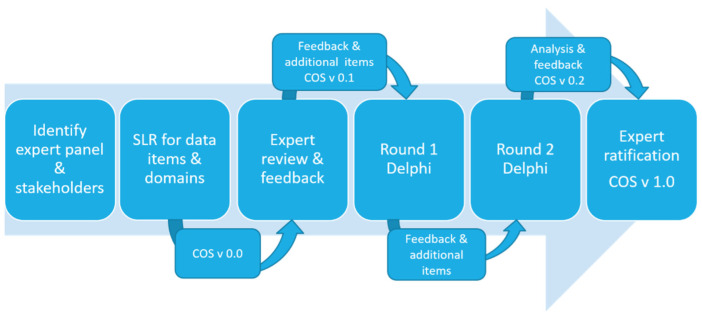
Methodology for Core Outcome Set identification and consensus process. SLR, systematic literature review; COS, core outcome set; v 0.0, version 0.0; v 0.1, version 0.1; v 0.2, version 0.2; v 1.0, version 1.0.

**Figure 2 ijerph-19-09666-f002:**
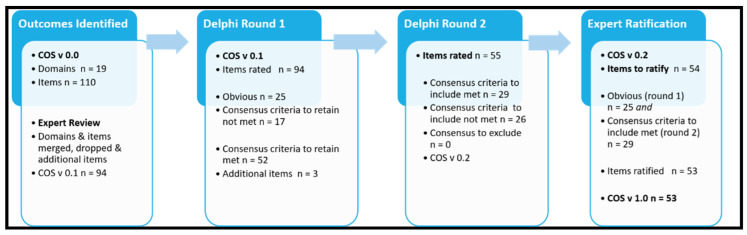
Identification of outcomes and consensus process flow. COS v = core outcome set and version 0.0, 0.1, 0.2 or 1.0.

**Table 1 ijerph-19-09666-t001:** Demographics of Delphi Consensus Participants.

	Round 1 Participants (n)	Round 2 Participants (n)
**COUNTRY**		
Australia	10	8
Canada	7	7
United Kingdom (UK)	7	7
Ireland	5	4
Brazil	4	2
United States of America (USA)	3	3
Netherlands	3	2
Belgium	2	2
Portugal	2	1
Unknown	2	1
France	1	1
Czech Republic	1	1
Argentina	1	1
Slovenia	1	1
Italy	1	0
**GENDER**		
Female	29	26
Male	19	13
Unspecified	2	2
**EXPERIENCE SB RESEARCH**		
<5 years	21	13
5–10 years	18	18
>10 years	11	10
**CURRENT ROLE**		
Academic	24	22
Researcher	11	9
Clinical	4	1
Academic; Researcher	6	5
Clinical; Academic; Researcher	3	3
Clinical; Academic	1	1
Other	1	0

**Table 2 ijerph-19-09666-t002:** Final Core Outcome Set of Descriptors and Outcomes for use in all Sedentary Behaviour Interventional Research.

Domain & Reporting Statistic	Outcome Name	Measure/Report	When to Measure	
Demographics ^1 or 2^	Age	years	baseline	
	Gender	male/female/other	baseline	
	Population type	healthy sedentary; healthy active; clinical cohort	baseline	
	BMI (body mass index)	$\frac{weight\;\left(\mathrm{kgs}\right)}{height\;{\left(m\right)}^{2}}$	baseline	follow-up
Device Details and Wear Time Criteria	Device type	accelerometer with inclinometry function		
	Device sensor position	placement of sensors on body		
	Device minimum wear time	hours/day		
	Device minimum wear time	days/week		
Device Wear Time Measured ^1^	Total daily wear-time	minutes or hours per day		
	Waking/non-sleep wear-time	minutes or hours per day, while awake		
	Sleep wear-time	minutes or hours per day, while sleeping		
Posture Related Outcomes ^1 & 3^	Sedentary time	minutes or hours per day	baseline	follow-up
		% daily waking hours	baseline	follow-up
		% of total daily hours	baseline	follow-up
	Sitting time	minutes or hours per day	baseline	follow-up
		% daily waking hours	baseline	follow-up
		% of total daily hours	baseline	follow-up
	Standing time	minutes or hours per day	baseline	follow-up
		% daily waking hours	baseline	follow-up
		% of total daily hours	baseline	follow-up
	Stepping time	minutes or hours per day	baseline	follow-up
		% daily waking hours	baseline	follow-up
		% of total daily hours	baseline	follow-up
Sedentary Breaks ^1&4^	Sedentary breaks; Sit to stand or upright transitions	n/day; number of sedentary breaks daily	baseline	follow-up
	Sedentary breaks; Movement breaks	n/day; number of sedentary breaks daily	baseline	follow-up
Sedentary Bouts ^1&3^	Prolonged sedentary bouts > 30 min	average duration of bout, mins	baseline	follow-up
		total duration of bouts, minutes or hours per day	baseline	follow-up
	Prolonged sedentary bouts > 60 min	average duration of bout, minutes	baseline	follow-up
		total duration of bouts, minutes or hours per day	baseline	follow-up
Physical Activity ^1&3^	Light intensity physical activity, time	minutes/day	baseline	follow-up
		% of daily waking hours	baseline	follow-up
	Moderate intensity physical activity, time	minutes/day	baseline	follow-up
		% of daily waking hours	baseline	follow-up
	Vigorous intensity physical activity, time	minutes/day	baseline	follow-up
		% of daily waking hours	baseline	follow-up
	Moderate–vigorous intensity physical activity, time	minutes/day	baseline	follow-up
		% of daily waking hours	baseline	follow-up

Reporting statistic for each item in the domain; ^1^ = mean, standard deviation, and range; ^2^ = number or % in each category; ^3^ = change in mean and confidence interval; ^4^ = change in mean and confidence interval or change in rate.

## Data Availability

All data are available in the Appendix A.

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
