# Peer review of "A Standardised Core Outcome Set for Measurement and Reporting Sedentary Behaviour Interventional Research: The CROSBI Consensus Study"

_ijerph, 2022, doi:10.3390/ijerph19159666_

Round 1

Reviewer 1 Report

Dear authors, 

Congratulation for your manuscript, it is novel in this field, but some minor changes are necessary:

- Abstract: Please, eliminate de abbreviations. 

- Title of the tables: Please, give more information about the results of each table of the manuscript.

Reviewer 2 Report

This research involved the development of a COS to guide measurement and reporting of descriptors and outcomes in SB intervention research in adult populations and provides a data collection and reporting tool for use across many adult populations and domains of living. This tool will help reduce the heterogeneity in SB research as there will be a core group of outcome measures. This is a timely and necessary study, and the results will likely have wide impact. 

The study was very well conceptualized and executed, with all steps in the process rationalized and followed through on with scientific rigour. 

My recommendation is a minor revision and to accept the manuscript as is otherwise. The last sentence in the abstract is not a complete sentence and needs to be edited. 

Reviewer 3 Report

Thank you very much for the opportunity to review this interesting manuscript. 

The present paper aims to identify and validate, by consensus, a core set of domains and items to report in intervention studies to reduce sedentary behaviours. The study objective is clearly defined and the study design appropriate. Sedentary behaviour has been extensively investigated by the existing scientific literature but there is still some confusion over the definitions and the best tools to use in different type of studies.

Therefore, the proposed consensus could be useful to help scientists and professionals finding the proper study methods and tools to use in their investigations and interventions.

The paper is well written and clear, however I have some comment that can improve its quality and facilitate the understandability.

Specific comments:

Line 2: cut “Title”

Title:

I would simplify it. I needed to read it three times before understanding the contents.

Abstract:

Lines 33-34 the sentence is probably part of the previous phrase or is missing some words.

Introduction:

The introduction is well written but I think gives few information about the importance of the topic and the need of this consensus while it focuses too much (in my opinion) on some methodological aspects that underlie the project without clear connections between paragraphs.

Line 50-56: some references are needed in the this paragraph

Methods:

Line 92: Error hyperlink

Lines 100-110: the description of the systematic literature review is not detailed enough (inclusion and exclusion criteria for example..). It is not clear if the authors carried out a systematic review or if they just used a previous published review in order to identify the outcomes (from figure 1 it seems that the authors carried out a SLR..in this case you should be more detailed).

Figures: the quality is low 

Line 116: I don’t think that authors initials are needed in the text. You can specific the contribution in the proper section at the end of the paper.

Lines 131-132: please better connect these two paragraphs rephrasing paragraph 132-136

Line 139: Error Hyperlink

Results:

Line 192: 59 participants. In my opinion, for a consensus, a bigger sample, more statistically representative, would have been needed. 

Line 246: the link of the website doesn’t bring to the COS user friendly format. I was not able to find it on the website.

Discussion:

I believe that a practical implication and application section is missing as well as some references to validated scales and tools that can help the reader to better understand how to use the consensus. Moreover, you should add some references at the beginning of the discussion and also line 291.
